# The Benefits to Bone Health in Children and Pre-School Children with Additional Exercise Interventions: A Systematic Review and Meta-Analysis

**DOI:** 10.3390/nu15010127

**Published:** 2022-12-27

**Authors:** Callum McCaskie, Aris Siafarikas, Jodie Cochrane Wilkie, Vanessa Sutton, Paola Chivers, Nicolas H. Hart, Myles C. Murphy

**Affiliations:** 1School of Medical and Health Sciences, Edith Cowan University, Joondalup, WA 6027, Australia; 2Department of Endocrinology and Diabetes, Perth Children’s Hospital, Nedlands, WA 6009, Australia; 3School of Medicine and Telethon Kids Institute, University of Western Australia, Crawley, WA 6009, Australia; 4Institute for Health Research, The University of Notre Dame Australia, Fremantle, WA 6160, Australia; 5Western Australian Bone Research Collaboration, Fremantle, WA 6160, Australia; 6Faculty of Health, Southern Cross University, Gold Coast, QLD 4225, Australia; 7Nutrition and Health Innovation Research Institute, School of Medical and Health Sciences, Edith Cowan University, Joondalup, WA 6027, Australia; 8Caring Futures Institute, College of Nursing and Health Science, Flinders University, Adelaide, SA 5042, Australia; 9School of Sport, Exercise and Rehabilitation, Faculty of Health, University of Technology Sydney, Moore Park, NSW 2007, Australia; 10School of Health Sciences and Physiotherapy, The University of Notre Dame Australia, Fremantle, WA 6160, Australia

**Keywords:** DXA, pQCT, impact exercise, jump

## Abstract

Objective: Determine if exercise interventions, beyond what is already provided to children and preschool children, improve bone health and reduce fracture incidence. Design: Systematic review and meta-analysis reported using the PRISMA guidelines. Certainty of evidence was assessed using GRADE recommendations. Data sources: Five electronic databases were searched for records: PUBMED; CINAHL; CENTRAL; SPORTDiscus; Web of Science. Eligibility criteria for selecting studies: Randomised, quasi-randomised and non-randomised controlled trials (including cluster-randomised) assessing the impact of additional exercise interventions (e.g., increased physical education classes or specific jumping programs) on bone health in children (6–12 years) and pre-school children (2–5 years) without dietary intervention. Results: Thirty-one records representing 16 distinct clinical trials were included. Dual-energy X-ray Absorptiometry (DXA) and/or peripheral Quantitative Computed Tomography (pQCT) were used to quantify bone health. Increased femoral neck bone mineral content in children with additional exercise interventions (*n* = 790, SMD = 0.55, 95% CI = 0.01 to 1.09) was reported, however this was not significant following sensitivity analysis. Other DXA and pQCT measures, as well as fracture incidence, did not appear to significantly differ over time between intervention and control groups. No studies reported adverse events. Studies failed to report all domains within the TIDieR checklist. All studies were at high risk of bias using the Cochrane RoB Tool 2.0. The certainty of the evidence was very low. Conclusions: The addition of exercise interventions, beyond what is provided to children, does not appear to improve DXA and pQCT measures of bone health. The effect of additional exercise interventions on bone health in pre-school children is largely unknown. Future trials should ensure adherence is clearly reported and controlled for within analysis as well as including reports of adverse events (e.g., apophysitis) that occur due to increased exercise interventions.

## 1. Introduction

It is widely agreed that bone mass accrual in childhood and adolescence determines peak bone mass and the risk of osteoporosis later in life [1,2]. Furthermore, the accrual of bone mass can be optimised by mechanical loading through exercise interventions [3,4,5,6] with bone adaptation most readily occurring with mechanotransduction of impact loading (e.g., running or jumping) [7,8]. Engagement in activities involving running and jumping is not only beneficial for bone, but also has wide ranging health benefits, including reducing obesity and improving psychological well-being [5].

The World Health Organisation recommends “children and young people aged 5–17 years old should accumulate at least 60 min of moderate to vigorous-intensity physical activity daily” [9]. However, a recent analysis from the Australian Institute for Health and Welfare showed that less than one-quarter of children aged 5–14 years meet this recommendation [10]. Worryingly, fracture rates within Australia have also been shown to have increased over time [11].

Given the importance of specific exercise to bone development, such as running or jumping, one of the simplest solutions would be, in theory, to increase exercise in children to optimise benefits from exercise starting early in life. If being specific to bone health, the exercise interventions would need to include specific aspects such as timing and duration of exercise, motivation and mode of activity [12]. Whilst the effect of exercise interventions for bone health in children have been quantified previously, new research is available. Furthermore, these previous systematic reviews contain methodological concerns. Behringer et al. 2014 pooled the results of all studies, irrespective of the region Bone Mineral Content (BMC) or areal bone mineral density (aBMD) was assessed, at their final follow-up with these time points ranging from 6–208 weeks, which given the time taken for bone to adapt following mechanical loading is a problem. Ref. [3] Tan et al. 2014 listed studies that had assessed the same data at different timepoints separately, influencing conclusions, as opposed to treating all published papers related to a single cohort as one trial (e.g., Malmo Paediatric Osteoporosis Prevention Study). Ref. [6] Nikander et al. 2010 also failed to delineate multiple studies using the same data and included both in pooled analysis [13,14], and presented pooled results within meta-analysis of outcomes assessing different regions (e.g., tibia and femoral measures were pooled).

Due to the sample sizes of previous meta-analyses being incorrectly inflated due to the inclusion of several published studies from a single trial or by pooling different body regions, the positive relationship between existing exercise regimes and bone health shown in previous reviews remains uncertain.

### Objectives

Our primary objective was to determine if additional exercise interventions, beyond what is already provided, in children and preschool children improve bone health. Our secondary objective was to determine if additional exercise interventions, beyond what is already provided, in children and preschool children reduce fracture incidence.

## 2. Materials and Methods

### 2.1. Guidelines

The systematic review and meta-analysis was designed and reported in accordance with the Preferred Reporting Items for Systematic Reviews and Meta-Analyses (PRISMA) [15].

### 2.2. Prospective Registration

This systematic review protocol has been registered with PROSPERO (CRD42022300902) on the 14^th^ May 2022 [https://www.crd.york.ac.uk/prospero/].

### 2.3. Data Management

All records were stored online using Covidence (Covidence systematic review software, Veritas Health Innovation, Melbourne, Australia). All extracted data were stored in a Microsoft Excel document, within Microsoft Teams.

### 2.4. Inclusion Criteria

#### 2.4.1. Participants

We included children (6–12 years) and preschool children (2–5 years) within this review who did not have medical comorbidities. We did not restrict inclusion based on body mass index.

#### 2.4.2. Primary Outcomes

The primary outcome of interest for this review was bone health. We included any measure of bone assessed using Dual-energy X-ray Absorptiometry (DXA) or peripheral Quantitative Computed Tomography (pQCT) with baseline and follow-up outcomes.

#### 2.4.3. Secondary Outcome

We included the incidence of fractures (occurrences) as a secondary outcome of this systematic review.

#### 2.4.4. Types of Studies

We included randomised, quasi-randomised and non-randomised clinical trials where at least one study arm used an exercise intervention for bone health and was compared to a control group (standard care/standard physical activity or other exercise intervention). Observational trials, case reports or series, clinical observations and systematic reviews were excluded.

### 2.5. Search Strategy

A single study author (MM) performed all search strategies from database inception to 3 May 2022 and downloaded all records into Endnote 20.4.1 (Clarivate Analytics, Philadelphia, PA, USA), and subsequently uploaded them into Covidence.

#### 2.5.1. Electronic Searches

Searches using free-text terms (Appendix A) were performed within the following electronic databases; PUBMED, CINAHL (Full-text), CENTRAL, SPORTDiscus, Web of Science. Limitations were used, where able, within individual databases with the complete search documentation provided within Appendix B.

#### 2.5.2. Searching Other Sources

Reference lists of reviews and retrieved articles were checked for additional studies missed in the electronic database search. The ePublication lists of key journals in the field were screened to identify studies that had yet to be indexed.

#### 2.5.3. Study Selection

Titles and abstracts of records identified by the search strategy were screened for their eligibility by two independent reviewers (MM/CM). Full-text screening was performed by the same two independent reviewers (MM/CM). Disagreements were resolved via consensus. The overall reasons for full-text articles being excluded are reported within the PRISMA flow chart [15].

### 2.6. Dealing with Multiple Records from the Same Cohort

Where multiple records were identified for a single cohort (as identified directly via manuscript text, referencing, clinical trial registration matching or ethical approval number matching) records were pooled to represent a single trial. Nine records were identified from the Swedish Malmo Paediatric Osteoporosis Prevention study [16,17,18,19,20,21,22,23,24]. Two records were identified from the Canadian AS!BC study [13,14]. Three records were identified from the Canadian Impact Loading study [25,26,27]. Three records were identified from the American Jumping Intervention study [28,29,30]. Two records were identified from the Swiss jumping study [31,32].

### 2.7. Dealing with Missing Data

Where the baseline or follow-up mean was not reported, the corresponding authors were contacted to provide these data directly. Ten studies did not provide a follow-up mean within their manuscript and were contacted for these data [18,19,24,26,32,33,34,35,36,37]. The corresponding author of two studies reported the data no longer exists [33,34]. Eight study authors did not respond to a request for data [18,19,24,26,32,35,36,37], however this was not unexpected with many studies being greater than ten years old. Where a standard deviation was not reported, yet the confidence interval was, a standard deviation was obtained using the methods described within the Cochrane Handbook for Systematic Reviews of Interventions [38]. Where no measure of deviation was provided, the standard deviation was imputed from a trial with an identical outcome measure at the closest timepoint, as recommended within the Cochrane Handbook for Systematic Reviews of Interventions [38].

### 2.8. Data Extraction

Data were abstracted from studies into Microsoft Excel independently by two review authors (CM/VS). Disagreements were resolved via group consensus with a senior reviewer (MM). The following information was abstracted from included studies: primary author; year of publication; country of origin; funding; competing interests; study design; study population; overall sample size; age range; details related to the intervention to complete the Template for Intervention Description and Replication (TIDieR) [39] checklist; mean (SD) age; mean (SD) weight; mean (SD) height; mean (SD) body mass index; participant sex; sample size at baseline and follow-up, mean (SD) of outcome variable (e.g., BMC) at baseline and follow-up; number of fractures.

### 2.9. Assessment of Methodological Quality

Two independent review authors (CM/AS) assessed the quality of included studies using the Cochrane Risk of Bias 2.0 Tool [40]. Disagreements were resolved via group consensus with a senior reviewer (MM). An overall judgement of methodological quality was assigned based on a ‘worst-item-counts’ basis with studies being assigned ‘high risk’ if at least one item was reported as “high risk”, unclear if no items are reported as ‘high risk’ and at least one item is reported as ‘unclear risk’, and ‘low risk’ if all items are reported as ‘low risk’ [41,42].

### 2.10. Assessment of Diversity and Heterogeneity

A Chi square test was used to evaluate statistical heterogeneity [38]. The I^2^ statistic informed between study heterogeneity based on the *p* value being <0.10 or the I^2^ value being >40% [38].

### 2.11. Assessment of Reporting Biases

The influence of small sample and publication bias were considered. Publication bias was not assessed using Egger’s test [43] due to the small number of studies (<10) included within each meta-analysis [38]. Alternatively, funnel plots were inspected when there were greater than five studies included within meta-analysis. We used the risk of bias criterion ‘sample size’ to account for small study biases, as per previous meta-analysis [44]. Where fewer than 50 participants were included, the study was classified as high-risk. Where between 50 and 200 participants were included, studies were classified as moderate risk and when greater than 200 participants were included studies were classified as low risk of small sample bias [45,46].

### 2.12. Data Synthesis

Data abstracted from included studies are presented within our Summary of Findings Table. The standardised mean difference (95% confidence intervals) was determined from pooled studies using random-effects, inverse variance models within Review Manager. Statistical significance was identified at *p*  <  0.05.

### 2.13. Sensitivity and Subgroup Analysis

We had planned to perform sub-group analysis, however based on the limited number of studies we did not proceed with this. We performed sensitivity analysis for meta-analyses consisting of three or more studies and demonstrating statistical heterogeneity with a clear outlier. Sensitivity analysis was conducted by excluding the outlier and exploring whether that altered the significance of the meta-analysis. With the limited number of included studies, and the lack of statistical heterogeneity observed within most meta-analyses, sensitivity analysis was not performed for every meta-analysis.

### 2.14. Assessment of the Certainty of the Body of Evidence

We assessed the certainty of the body of evidence using the GRADE approach (as recommended in the Cochrane Handbook for Systematic Reviews of Interventions) [47]. This process involved overall judgements on the quality of the body of evidence based on the overall risk of bias, inconsistency, indirectness, imprecision and publication bias.

### 2.15. Deviations to Protocol

We had planned to extract components of included study interventions using the Consensus on Exercise Reporting Template (CERT) framework [48], however due to the broader nature of the interventions prescribed within included studies this was replaced with the TIDieR checklist.

## 3. Results

### 3.1. Selection of Studies

We identified 477 records following the removal of duplicates within Covidence. After title and abstract screening, 51 records proceeding to full-text review with 31 records being included within this review [13,14,16,17,18,19,20,21,22,23,24,25,26,27,28,29,30,31,32,33,34,35,36,37,49,50,51,52,53,54,55]. These 31 records represented 16 distinct clinical trials (e.g., 15 studies were follow-up publications from the original study publication). Study inclusion is demonstrated within our PRISMA flow chart (Figure 1) and a study-by-study list of exclusion is provided within Appendix C.

### 3.2. Study Information

Full study data are presented within Table 1, with funding information presented within Appendix D. Of the sixteen distinct trials, 15 were randomised [13,16,25,28,32,34,35,36,37,49,50,51,53,54,55] and one was not [52]. Of the 15 randomised trials, four were individually randomised [28,36,37,50] and eleven were cluster randomised [13,16,25,30,32,35,49,51,53,54,55]. Fifteen trials included children [13,16,25,28,32,34,35,37,49,50,51,52,53,54,55] and one trial included pre-school children [36]. One trial exclusively included overweight or obese children [37]. Trials were performed in a variety of countries: Five in the United States of America [28,34,36,37,50]; three in Canada [13,25,53]; two in Australia [51,55]; two in Denmark [35,52]; two in Switzerland [32,49]; one in South Africa [54]; and one in Sweden [16].

### 3.3. Intervention Information

The TIDieR checklist was completed to detail all information related to the interventions provided within each included study, and is presented within Appendix E. All trials compared additional exercise interventions to standard exercise in children and pre-school children and most (11/16) trials used additional exercises specifically selected for bone adaptation, predominantly based around additional jumping exercise [13,16,25,28,32,35,36,37,49,53,55]. The duration of additional exercise interventions varied from 20 weeks to four years between studies, with all additional exercise interventions being performed face-to-face. Little equipment was needed for most studies and the interventions typically took place at school.

### 3.4. Effect of Exercise Interventions on Dual-Energy X-ray Absorptiometry Measures

The complete dataset for DXA outcomes are presented within Appendix F and all studies reported below assessed children, not pre-school aged children.

#### 3.4.1. Bone Mineral Content

Meta-analysis was possible for BMC of the whole body, femoral neck, and lumbar spine (Figure 2). Only short-term femoral neck BMC was significantly improved with additional exercise interventions, when compared to standard exercise (SMD = 0.55, 95% CI 0.01 to 1.09, *p* < 0.05). Sensitivity analysis performed on short-term femoral neck BMC by excluding Fuchs et al. 2001 as an outlier made this result non-significant. No other measure of BMC was significantly changed with additional exercise interventions.

#### 3.4.2. Areal Bone Mineral Density

Meta-analysis was possible for aBMD of the whole body, femoral neck and lumbar spine (Figure 3). No significant differences were detected with additional exercise interventions, when compared to standard exercise. Sensitivity analysis performed on short-term whole body and femoral neck aBMD by excluding McKay et al. 2005, which resolved heterogeneity and the results remained not significant.

#### 3.4.3. Bone Cross-Sectional Area

Meta-analysis was possible for cross-sectional area (CSA) of the femoral neck for outcomes less than 12 months (Appendix G). No significant differences were seen with additional exercise interventions, when compared to standard exercise at less than 12 months within the femoral neck. In additional to the meta-analysis, individual studies that measured CSA within the femoral neck at eight years and the lumbar spine at 6 months did not demonstrate significant differences between groups [16,17,28].

#### 3.4.4. Bone Area and Width

Meta-analysis was not possible for any measures of bone area or width, with no significant differences being reported within the study that assessed these measures [23,27].

### 3.5. Effect of Exercise Interventions on Peripheral Quantitative Computed Tomography Measures

The complete dataset for pQCT outcomes is presented within Appendix H and all studies reported below assessed children, not pre-school aged children.

#### 3.5.1. Volumetric Bone Mineral Content

Meta-analysis was possible for 4% tibial volumetric bone mineral content (vBMC) with no significant differences seen with additional exercise interventions, when compared to standard exercise (Appendix I). Meta-analysis was not possible for other measures of volumetric bone mineral content (vBMC), with no significant differences being reported within the one study that observed these measures at different sites of the tibia (14%, 38%, and 66%) [49].

#### 3.5.2. Volumetric Bone Mineral Density

Meta-analysis was possible for tibial vBMD at 4% and 38% sites with no significant differences seen with additional exercise interventions, when compared to standard exercise (Figure 4). Individual studies of the tibia (4%, 8%, 14%, 35%, 50%, 66%) and radius (38%) also did not observe significant between-group differences [13,49,51,54,55].

#### 3.5.3. Trabecular Bone Mineral Density

Meta-analysis was possible for 4% tibial trabecular vBMD with no significant differences seen with additional exercise interventions, when compared to standard exercise (Appendix J). In additional to the meta-analysis, one study of the radius (4%) did not observe significant between-group differences [54,55].

#### 3.5.4. Bone Cross-Sectional Area

Meta-analysis was possible for tibial CSA at 4% and 38% sites with no significant differences seen with additional exercise interventions, when compared to standard exercise (Appendix K). Individual studies of the tibia (8%, 14%, 50%, 66%) did not observe significant between-group differences [13,49,51,54].

#### 3.5.5. Total Cortical Area

Meta-analysis was not possible for any measures of total cortical area of the tibia (38%, 50%, 66%), with no significant differences being observed within the studies that reported these measures [13,51,54].

#### 3.5.6. Total Cortical Thickness

Meta-analysis was not possible for any measures of total cortical thickness of the tibia (38%, 66%), with no significant differences being observed between studies that reported these measures [51,54].

#### 3.5.7. Stress–Strain Index

Meta-analysis was possible for 38% tibial polar stress–strain index (SSIPOL) with no significant differences seen with additional exercise interventions, when compared to standard exercise (Appendix L). Individual studies of the tibia (14%, 50%, 66%) did not observe significant differences [13,49,51,54].

#### 3.5.8. Bone Strength Index

Meta-analysis was not possible for the tibial bone strength index (4%, 8%), with no significant differences being observed within the studies that reported these measures differences [13,54].

#### 3.5.9. Other

Meta-analysis was not possible for tibial endosteal circumference or periosteal circumference (38%). No significant differences were observed within the studies that reported these measures [54,55].

### 3.6. Fracture Incidence

Fracture incidence was reported in children, not preschool children, within a single trial, with four publications reporting incidence at different timepoints (Appendix M) and no significant differences at any follow-up time point (Figure 5) [18,19,20,24].

### 3.7. Assessment of Quality in Included Studies

All studies were judged to be at high-risk of bias with the overall risk of bias judgements presented within Appendix N. Risk of bias arising from the randomisation process was low in 88% of included studies. Risk of bias arising from the timing of identification or recruitment of participants into the randomised trial was low in all studies (100%). No studies reported a low risk of bias due to deviations from the intended interventions (effect of assignment to intervention). One study was judged as low risk (6%) for risk of bias due to deviations from the intended interventions (effect of adhering to intervention). Risk of bias due to missing outcome data was low for 94% of studies. All studies were low risk (100%) for risk of bias in measurement of the outcome and risk of bias in selection of the reported result. Finally, funnel plots were inspected for all meta-analyses with greater than five included studies and did not indicate any evidence of publication bias.

### 3.8. Assessment of the Certainty of the Body of Evidence

This systematic review and meta-analysis included randomised and non-randomised controlled clinical trials, with the initial level of evidence (before GRADE judgements) classified as high. Study limitations, from risk of bias, downgraded the certainty of the evidence once. As most meta-analyses reported the majority of individual study confidence intervals crossing zero, minimal heterogeneity was seen between studies, and if present could be resolved by the exclusion of a single study. Thus, the certainty was not downgraded for inconsistency. Trials included similar populations, interventions and outcome measures, so the certainty of the evidence was not downgraded for indirectness. Trials included samples unlikely to be large enough to detect between group effect sizes and the certainty of the evidence was downgraded two levels for imprecision. There was no publication bias detected via funnel plots and the certainty of the evidence was not downgraded for publication bias. The overall certainty of the evidence was judged to be very low, meaning little confidence exists in the reported effect sizes and that the true effects are likely to be substantially different from those reported.

## 4. Discussion

This systematic review and meta-analysis analysis demonstrated that the addition of existing exercise interventions in children did not result in significant improvements to bone health. This review was only able to identify a single study that assessed the effect on bone health in pre-school children (aged between 2–5 years) and they did not report sufficient data for analysis. Finally, due to risk of bias due to deviations from the intended interventions and measurement of outcomes, all studies were at high risk of bias and the overall quality of the evidence was very low.

Femoral neck BMC significantly improved at <15-month follow-up with additional exercise interventions in children, however this result did not remain significant following sensitivity analysis. Whole-body and lumbar spine BMC did not significantly change at any follow-up timepoint with the additional exercise interventions. Additional exercise interventions also failed to significantly improve aBMD at the femoral neck, lumbar spine or whole-body.

No significant improvements were observed in tibial vBMD at any level with additional exercise interventions. However, vBMD may not be an overly specific measure and has been shown to be unable to differentiate between female and male Australian Football players [56]. Despite significant differences between Australian Football Men’s and Women’s players according to other musculoskeletal morphological characteristics (e.g., bone mass, cross-sectional area and robustness), vBMD (measured via pQCT) was unable to differentiate between cohorts. Furthermore, despite being one of the universal measures of hard-tissue health, BMD may have limited utility in evaluating bone health and strength [57].

A number of additional measures of bone strength were assessed using pQCT. No significant improvements were observed in trabecular vBMD, CSA, cortical area, cortical thickness or SSIPOL with additional exercise interventions. This was a relatively unexpected finding as longitudinal exercise has resulted in favourable hard-tissue improvements in adult athletes previously [56,58]. Specifically, elite male athletes exhibit superior hard-tissue characteristics in their support leg versus their kicking leg, which is likely due to the different loading patterns between limbs [56]. The support limb typically experiences more frequent impacts as it acts as the stabilising leg during kicking actions and is typically the dominant single-leg jumping and landing leg [58]. Furthermore, greater hard-tissue asymmetry between kicking and support limbs was found in experienced athletes over younger inexperienced athletes, further illustrating the positive effect of longitudinal specific exercise loading on bone health. However, the addition of exercise interventions in children within this systematic review did not result in favourable improvements in hard-tissue characteristics.

These findings suggest, at least in children aged 6–12 years, that either normal growth and development, or standard exercise associated with school and childhood may be sufficient to increase the hard-tissue characteristics of bone, as significant within-group improvements over time were observed in both intervention and control groups in all studies. It may be that the magnitude of change in hard-tissue characteristics through standard exercise and normal human physical development was adequate and any additional exercise interventions could not produce any further improvements in hard-tissue as the physiological plateau was already reached. Alternatively, another explanation is that the additional exercise interventions were insufficient in producing positive osteogenic effects beyond what children were already being exposed to in standard exercise programs. However, the lack of data in children aged 2–5 years was unexpected as this is a group that have not yet commenced formal exercise interventions within school.

The results of our review do not align with the conclusions of previous systematic reviews in this field, with sharp contrasts in the findings of our meta-analyses. One reason might be that the meta-analysis performed by Behringer et al. 2014 [3] pooled all studies that measured bone health (e.g., pooled different anatomical locations), with follow-up ranging from 6–208 weeks, creating a much larger sample (2985 participants including 1640 intervention participants) providing more statistical power to analysis, whereas our largest meta-analysis contained 915 total participants. However, we feel that breaking up timepoints and regions as we have in this review, provides a far more accurate representation of the outcomes from exercise interventions.

No difference in fracture incidence was observed with additional exercise interventions in children in the single trial that recorded fracture incidence. This may be due to the lack of observable increases in bone health observed within our meta-analysis. Alternatively, other measures of bone strength not measured by either DXA or pQCT may have a greater link with fracture incidence. The use of nuclear magnetic resonance spectroscopy, which provides a far more detailed overview of bone structure and quality when compared to standard DXA or pQCT, might be informative in future studies. Alternatively, high-resolution pQCT (HR-pQCT) measures of hard-tissue characteristics were found to predict fracture in two separate systematic reviews [59,60]. However, prediction of fracture was observed to be stronger in the radius, rather than the tibia [59]. This may be problematic as radius fracture is likely the result of contact/impact, rather than overuse, which is distinct from the tibia, being a weight-bearing bone. It must also be noted that the lack of fracture incidence in children undertaking standard exercise may be due to the increased exercise levels being inherently associated with increased exposure to potentially risky physical activities (e.g., basketball), placing the children undertaking additional exercise interventions at higher risk of fracture.

With most studies at high risk of bias due to deviations from the intended interventions, future research should attempt to design and report their interventions in accordance with the TIDieR [39] and/or CERT checklists [48]. This will increase transparency [61,62] and ensure any tailoring or modifications to interventions are planned, recorded and accounted for within any analysis. Furthermore, adherence, including adherence to the control group protocol, was not recorded in all studies or factored into analysis, and this may have influenced outcomes [61,62]. In addition to adherence to the planned intervention, general physical activity should be considered to influence the results and future studies using activity trackers might provide greater insight into the role of exercise interventions for bone health.

Adverse events are an essential reporting element of clinical trials [63] but were not reported within included studies. Subsequently, not allowing us to discuss any potential adverse events associated with additional exercise interventions to inform discussion on the benefits of its implementation. One of the main drivers of apophysitis or bone stress injury is an overload of impact exercise [64]. Therefore, studies that increase exercise levels beyond standard physical activity should report adverse events. For example, if a future study was to demonstrate a small effect for additional exercise interventions in bone health, the benefits of implementing this study need to balance how many participants presented with injuries, such as apophysitis, as a direct result of the intervention to determine if it is worthwhile.

### Limitations

Several studies did not respond to requests for missing data or advised that data was no longer available, which was expected due to the age of many included studies, with measures of variability being inputted when data was missing as per Cochrane recommendations [38].

Unfortunately, not all studies provided separate outcomes by participant sex, which is important due to bone growth rate differences. However, given all but one study was randomized (and the non-randomized study provided separate data for boys and girls) we would not expect this to have influenced the results.

The DXA and pQCT procedures were not consistently reported between studies, and some differences did exist in assessment methodologies. However, given our meta-analysis included between-group differences (with the procedure for all groups within an individual study being the same), we would not expect this to have influenced the results.

## 5. Conclusions

The addition of exercise interventions, beyond what is provided to children, does not appear to improve DXA and pQCT measures of bone health. The effect of additional exercise interventions on bone health in pre-school children is largely unknown. Future trials should ensure adherence is clearly reported and controlled for within analysis as well as including reports of adverse events (e.g., apophysitis) that occurs due to increased exercise interventions.

## Figures and Tables

**Figure 1 nutrients-15-00127-f001:**
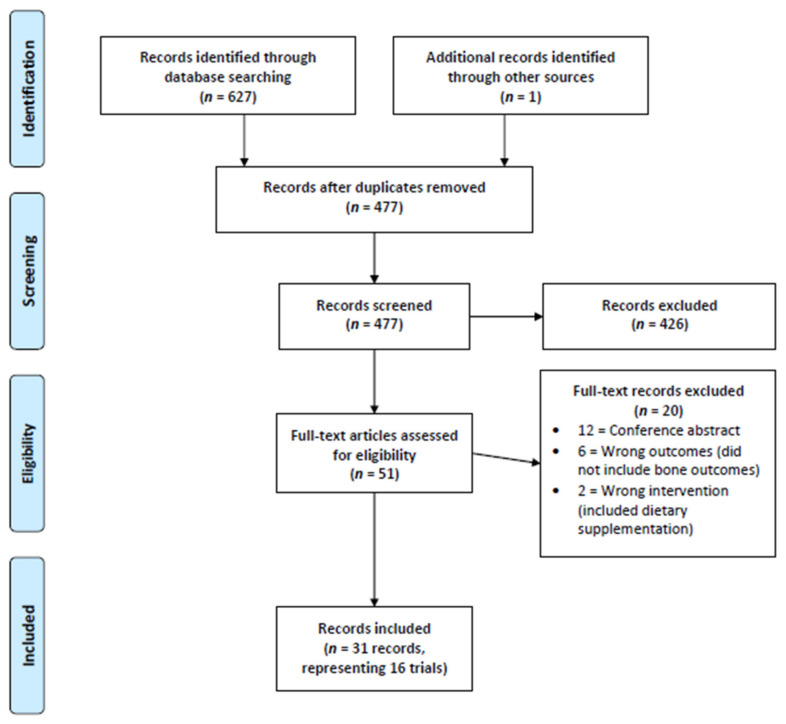
PRISMA Flow Chart.

**Figure 2 nutrients-15-00127-f002:**
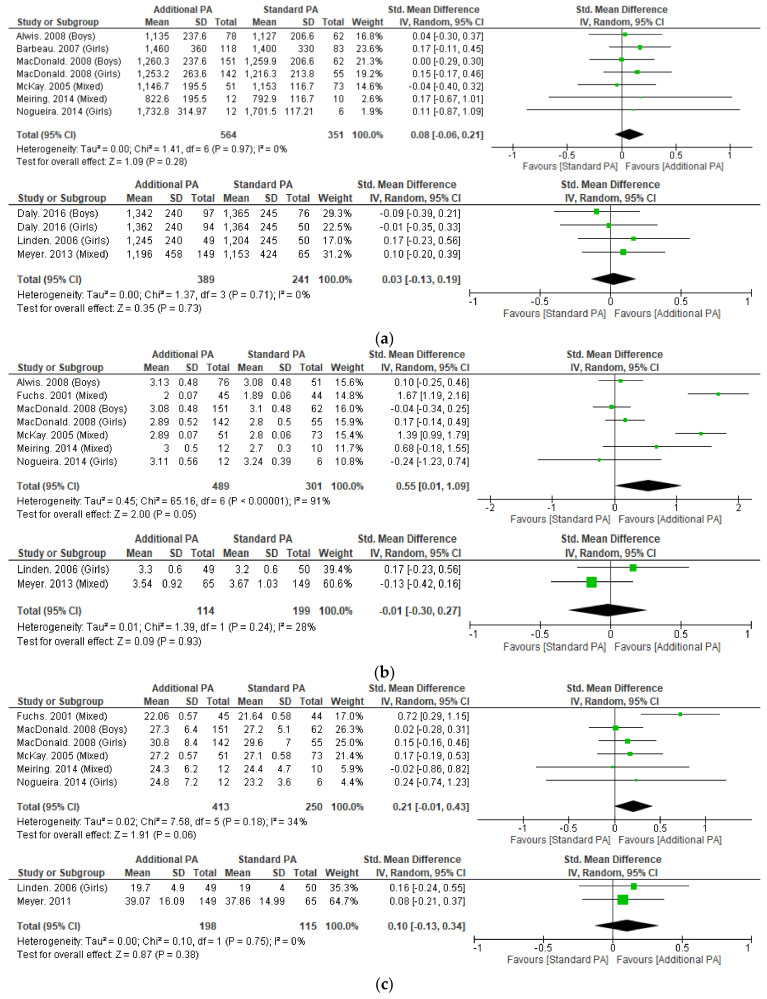
(**a**) Effect of additional exercise interventions on whole body bone mineral content. (**Top**) Less than 15 months follow-up, (**Bottom**) 24–48 months follow-up. Legend: PA = Physical Activity, Std = standardised, IV = inverse variance, SD- standard deviation, CI = confidence interval. (**b**) Effect of additional exercise interventions on femoral neck bone mineral content. (**Top**) Less than 15 months follow-up, (**Bottom**) 24–48 months follow-up. Legend: PA = Physical Activity, Std = standardised, IV = inverse variance, SD- standard deviation, CI = confidence interval. (**c**) Effect of additional exercise interventions on lumbar spine bone mineral content. (**Top**) Less than 15 months follow-up, (**Bottom**) 24–48 months follow-up. Legend: PA = Physical Activity, Std = standardised, IV = inverse variance, SD- standard deviation, CI = confidence interval.

**Figure 3 nutrients-15-00127-f003:**
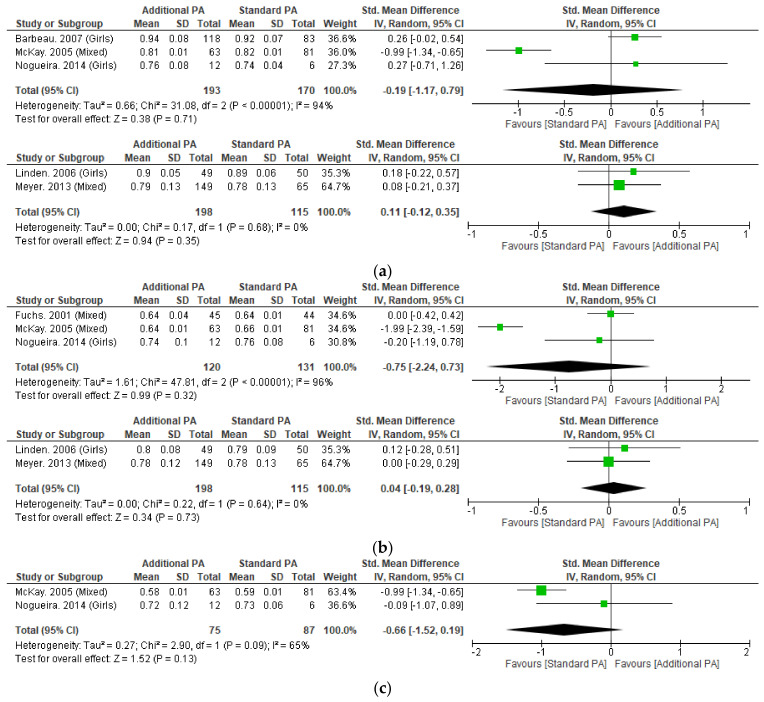
(**a**) Effect of additional exercise interventions on whole body areal bone mineral density. (**Top**) Less than 15 months follow-up, (**Bottom**) 24–48 months follow-up. Legend: PA = Physical Activity, Std = standardised, IV = inverse variance, SD- standard deviation, CI = confidence interval. (**b**) Effect of additional exercise interventions on femoral neck areal bone mineral density. (**Top**) Less than 15 months follow-up, (**Bottom**) 24–48 months follow-up. Legend: PA = Physical Activity, Std = standardised, IV = inverse variance, SD- standard deviation, CI = confidence interval. (**c**). Effect of additional exercise interventions on lumbar spine areal bone mineral density at less than 15 months follow-up. Legend: PA = Physical Activity, Std = standardised, IV = inverse variance, SD- standard deviation, CI = confidence interval.

**Figure 4 nutrients-15-00127-f004:**
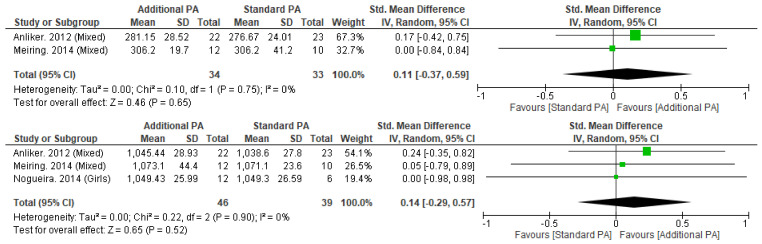
Effect of additional exercise interventions on volumetric bone mineral density at less than 12 months follow-up measures. (**Top**) 4% Tibia, (**Bottom**) 38% Tibia. Legend: PA = Physical Activity, Std = standardised, IV = inverse variance, SD- standard deviation, CI = confidence interval.

**Figure 5 nutrients-15-00127-f005:**
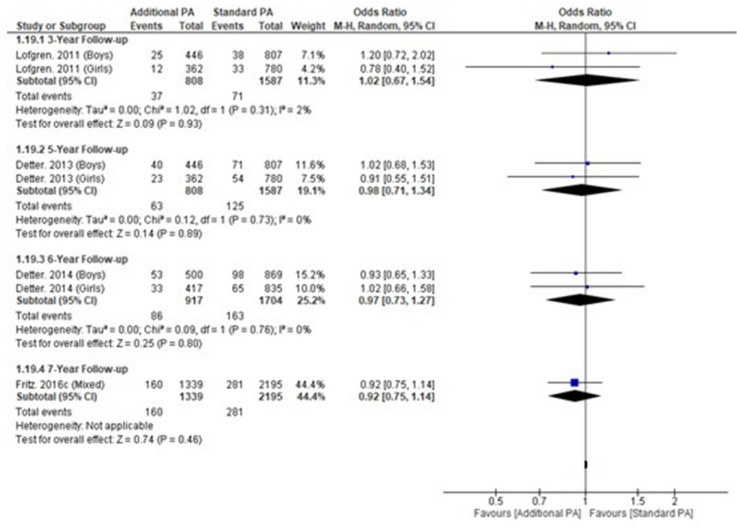
Fracture incidence within a single longitudinal trial over time.

**Table 1 nutrients-15-00127-t001:** Summary of included studies.

Study	Location	Design	Population	Sample Size (*n*)	Ages	Intervention Type	Group	Age, Mean (SD) Years	Height, Mean (SD) cm	Weight, Mean (SD) kg
Alwis. 2008	Sweden	Cluster RCT	Healthy	103	Children	Specific impact loading	INT	7.8 (0.6)	129.4 (6.5)	28.4 (5.8)
						Standard physical activity	CON	8.0 (0.6)	130.1 (6.7)	27.6 (5.4)
Anliker. 2012	Switzerland	Cluster RCT	Healthy	45	Children	Specific impact loading	INT	10.5 (1.2)	140 (12.0)	34.6 (7.7)
						Standard physical activity	CON	10.8 (1.1)	143 (7.0)	34 (5.7)
Barbeau. 2007	United States of America	RCT	Healthy	201	Children	General Physical Activity	INT	9.5 (NR)	NR	NR
						Standard physical activity	CON	9.5 (NR)	NR	NR
Daly. 2016	Australia	Cluster RCT	Healthy	727	Children	General Physical Activity	INT (Girls)	8.1 (0.3)	128.4 (5.5)	28.6 (5.9)
						General Physical Activity	INT (Boys)	8.1 (0.4)	130.4 (5.5)	28.9 (5.2)
						Standard physical activity	CON (Girls)	8.1 (0.4)	129 (5.3)	28.9 (5.7)
						Standard physical activity	CON (Boys)	8.2 (0.3)	129.9 (5.8)	28.7 (5.3)
Fuchs. 2001	United States of America	RCT	Healthy	89	Children	Specific impact loading	INT	7.5 (0.2)	125.1 (1.3)	27.1 (0.8)
						Standard physical activity	CON	7.6 (0.2)	126.8 (1.2)	28.0 (1.0)
Gutin. 2008	United States of America	Cluster RCT	Healthy	617	Children	General Physical Activity	INT	NR	NR	NR
						Standard physical activity	CON	NR	NR	NR
Hasselstrom. 2008	Denmark	Non-RCT	Healthy	704	Children	General Physical Activity	INT (Girls)	6.7 (0.3)	121.5 (4.9)	23.6 (3.2)
						General Physical Activity	INT (Boys)	6.8 (0.4)	124.2 (4.5)	24.4 (3.0)
						Standard physical activity	CON (Girls)	6.7 (0.4)	122.5 (4.6)	23.9 (3.8)
						Standard physical activity	CON (Boys)	6.8 (0.4)	123.6 (5.2)	24.7 (3.6)
Larsen. 2016	Denmark	Cluster RCT	Healthy	295	Children	Team Impact Sport	INT A (Girls)	9.3 (0.3)	137.6 (7.1)	33.0 (8.2)
						Team Impact Sport	INT A (Boys)	9.3 (0.4)	139.2 (6.4)	32.6 (5.4)
						Specific impact loading	INT B (Girls)	9.3 (0.3)	136 (5.7)	32.2 (6.7)
						Specific impact loading	INT B (Boys)	9.2 (0.4)	138.7 (5.5)	32.2 (7.3)
						Standard physical activity	CON (Girls)	9.4 (0.3)	138.7 (6.5)	33.5 (6.7)
						Standard physical activity	CON (Boys)	9.3 (0.3)	137.9 (5.3)	31.7 (5.2)
Macdonald. 2007	Canada	Cluster RCT	Healthy	410	Children	Specific impact loading	INT (Girls)	10.2 (0.6)	141.5 (7.5)	36.3 (8.4)
						Specific impact loading	INT (Boys)	10.2 (0.6)	141.5 (7.2)	37.2 (9.3)
						Standard physical activity	CON (Girls)	10.3 (0.5)	140.2 (7.5)	35.2 (8.7)
						Standard physical activity	CON (Boys)	10.3 (0.6)	141.2 (6.8)	39.7 (9.6)
MacKelvie. 2001	Canada	Cluster RCT	Healthy	198	Children	Specific impact loading	INT (Pre-pubertal Girls)	10.0 (0.6)	138.6 (7.6)	31.2 (6.1)
						Specific impact loading	INT (Early-pubertal Girls)	10.4 (0.7)	143.8 (7.7)	39.1 (8.3)
						Specific impact loading	INT (Boys)	10.2 (0.6)	140.6 (6.0)	35.5 (8.3)
						Standard physical activity	CON (Pre-pubertal Girls)	10.1 (0.5)	137.3 (6.2)	31.1 (5.6)
						Standard physical activity	CON (Early-pubertal Girls)	10.5 (0.6)	145.6 (6.4)	41.3 (8.3)
						Standard physical activity	CON (Boys)	10.3 (0.7)	141.8 (7.1)	36.6 (10.1)
McKay. 2000	Canada	Cluster RCT	Healthy	144	Children	Specific impact loading	INT	NR	133.9 (0.7)	30.5 (0.8)
						Standard physical activity	CON	NR	135.1 (1.1)	30.8 (1.0)
Meiring. 2014	South Africa	Cluster RCT	Healthy	22	Children	Standard physical activity	CON	9.3 (0.9)	135.1 (8.2)	30.6 (4.7)
						Specific impact loading	INT	9.7 (1.2)	135.9 (8.7)	30.0 (5.1)
Meyer. 2011	Switzerland	Cluster RCT	Healthy	291	Children	Specific impact loading	INT	8.8 (2.1)	133.3 (13.1)	30.7 (8.7)
						Standard physical activity	CON	8.8 (2.2)	134.2 (14.2)	30.4 (9.8)
Nogueira. 2014	Australia	Cluster RCT	Healthy	138	Children	Specific impact loading	INT	10.5 (0.6)	144.2 (6.7)	39.3 (9.4)
						Standard physical activity	CON	10.7 (0.6)	142.5 (7.1)	37.2 (7.2)
Specker. 2004	United States of America	RCT	Healthy	161	Preschool Children	Specific impact loading	INT	3.8 (0.5)	100.6 (6.1)	16.3 (2.2)
						Standard physical activity	CON	4.0 (0.6)	102.4 (5.4)	16.9 (2.3)
Staiano. 2018	United States of America	RCT	Overweight/Obese	46	Children	Specific impact loading	INT	NR	NR	NR
						Standard physical activity	CON	NR	NR	NR

Legend: RCT = randomised controlled trial, Children = 6–12 years, Preschool Children = 2–5 years, MVPA = Moderate to vigorous physical activity, POP = Pediatric Osteoporosis Prevention, INT = Intervention. CON = Control, NR = not reported.

## Data Availability

The complete dataset for this review is presented across the tables and appendices.

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
