# Peer review of "The Benefits to Bone Health in Children and Pre-School Children with Additional Exercise Interventions: A Systematic Review and Meta-Analysis"

_nutrients, 2022, doi:10.3390/nu15010127_

Round 1

Reviewer 1 Report

This paper is an original paper on the role of additional physical activity on bone health in children, with a systematic review coupled to a meta-analysis.
The methodology is well described and the studies are well presented in the appendix. The meta-analysis is well conducted.
The results are not in accordance with previous publications but that one is more rigorous with a good account of the bias.
However, the discussion analyses the results and the absence of effect of additional exercise interventions at this age.
The quality of the figures and characters for the meta-analysis must be improved.

Author Response

Thank you for your review and support for our manuscript. We have made the suggested amendments and provided a point-by-point response to all queries below.

  1. The quality of the figures and characters for the meta-analysis must be improved.
    1. Thank you, the figures are directly exported from Review Manager. Therefore, we have converted all figures to a 300DPI and provided these to Nutrients to improve the quality of the figures.

Reviewer 2 Report

There is an objective need for analysis such as the one presented by the authors to understand the correlation between physical activity and osteoporosis. Above all, the analysis reported by the authors is to be appreciated because it considers children. However, there are limitations in the work which also explain the conclusions of the paper, which the authors should discuss in the "discussion" section.

Here are the authors' conclusions:

“The addition of exercise interventions, beyond what is provided to children, does not appear to improve DXA and pQCT measures of bone health. The effect of additional exercise interventions on bone health in preschool children is largely unknown. Future trials should ensure adherence is clearly reported and controlled for within analysis as well as including reports of adverse events (e.g., apophysitis) that occur (please modify, not occurs) due to increased exercise interventions. “

In my opinion, the authors failed to observe differences between DXA and pQCT results for the following reasons which the authors should mention and discuss in the manuscript:

- DXA x-rays are also attenuated by the amount of fatty bone marrow as well as the bone. Was the DXA data adjusted for the amount of fat? Please see: Effect of increasing vertebral marrow fat content on BMD measurement, T-score status and fracture risk prediction by DXA by G.M. Blake, J.F. Griffith, D.K.W. Yeung, P.C. Leung, I. Fogelman. Bone, 2009;44, 495-501.

- male and female bone grow at different rates and reach different bone masses. Therefore the study had to be carried out on male and female samples separately

- The use of techniques such as DXA and pQCT can only estimate BMD but do not tell much about "bone quality" ( please see : Advanced imaging assessment of bone quality by H.K. Genant, Y. Jiang. Ann N Y Acad Sci, 2006;1068, 410-428)

 Bone is made up of bone minerals and bone marrow which in turn is made up mainly of water and fatty acids. In recent years, literature has highlighted how to evaluate bone quality it is necessary to analyze the bone as a whole, evaluating not only the BMD but also the relative quantity of water and fat in the bone marrow. To do this, nuclear magnetic resonance spectroscopy, MRS (please see ref. Behavior during aging of bone-marrow fatty-acids profile in women's calcaneus to search for early potential osteoporotic biomarkers: a 1H-MR Spectroscopy study by D Mattioli, V Vinicola, M Aragona, M Montuori, U Tarantino, S Capuani. Bone 2022; 164, 116514.  Water diffusion in cancellous bone by S Capuani Microporous and Mesoporous Materials 2013;178, 34-38).

Author Response

Thank you for your review and support for our manuscript. We have made the suggested amendments and provided a point-by-point response to all queries below.

  1. DXA x-rays are also attenuated by the amount of fatty bone marrow as well as the bone. Was the DXA data adjusted for the amount of fat? Please see: Effect of increasing vertebral marrow fat content on BMD measurement, T-score status and fracture risk prediction by DXA by G.M. Blake, J.F. Griffith, D.K.W. Yeung, P.C. Leung, I. Fogelman. Bone, 2009;44, 495-501.
    1. Included studies did not consistently report their DXA methodologies, and differences did exist between studies. However, data were not adjusted, that we could find, for the amount of fatty bone marrow. We recognise that this may be a limitation and have included it. However, we feel that the likelihood of this influencing outcomes is extremely unlikely as randomisation should account for between group differences in fatty bone marrow levels.

“The DXA and pQCT procedures were not consistently reported between studies, and some differences did exist in assessment methodologies. However, given our me-ta-analysis included between-group differences (with the procedure for all groups within an individual study being the same), we would not expect this to have influenced the results.”

  1. Male and female bone grow at different rates and reach different bone masses. Therefore, the study had to be carried out on male and female samples separately
    1. Thank you for this comment and we agree it is important to recognise sex differences. We have included that not all studies provided separate data for boys and girls, with the caveat that if studies were truly randomised this should be accounted for within analysis.

“Unfortunately, not all studies provided separate outcomes by participant sex, which is important due to bone growth rate differences. However, given all but one study was randomized (and the non-randomized study provided separate data for boys and girls) we would not expect this to have influenced the results.”

  1. The use of techniques such as DXA and pQCT can only estimate BMD but do not tell much about "bone quality" ( please see : Advanced imaging assessment of bone quality by H.K. Genant, Y. Jiang. Ann N Y Acad Sci, 2006;1068, 410-428)
    1. Whilst we agree that DXA does little to inform the quality of the bone, we do feel that pQCT, and its data that allow for more than just an estimation of BMD (e.g., polar stress strain index), do provide a more informative metric of bone quality. We do already provide discussion on the limitations of DXA and pQCT within the discussion and feel this is already appropriate.
  2. Bone is made up of bone minerals and bone marrow which in turn is made up mainly of water and fatty acids. In recent years, literature has highlighted how to evaluate bone quality it is necessary to analyze the bone as a whole, evaluating not only the BMD but also the relative quantity of water and fat in the bone marrow. To do this, nuclear magnetic resonance spectroscopy, MRS (please see ref. Behavior during aging of bone-marrow fatty-acids profile in women's calcaneus to search for early potential osteoporotic biomarkers: a 1H-MR Spectroscopy study by D Mattioli, V Vinicola, M Aragona, M Montuori, U Tarantino, S Capuani. Bone 2022; 164, 116514. Water diffusion in cancellous bone by S Capuani Microporous and Mesoporous Materials 2013;178, 34-38).
    1. The suggestion to highlight the benefits of MRS and include it in subsequent studies is excellent and we have amended our discussion to reflect this.

“The use of nuclear magnetic resonance spectroscopy, which provides a far more detailed overview of bone structure and quality when compared to standard DXA or pQCT, might be informative in future studies.”